# A Method to Develop the Driver-Adaptive Lane-Keeping Assistance System Based on Real Driver Preferences

**DOI:** 10.3390/s24051666

**Published:** 2024-03-04

**Authors:** Jiachen Chen, Hui Chen, Xiaoming Lan, Bin Zhong, Wei Ran

**Affiliations:** School of Automotive Studies, Tongji University, Shanghai 201804, China; 1410794@tongji.edu.cn (J.C.);

**Keywords:** lane-keeping assistance system, driver adaption, subjective and objective evaluation, naturalistic driving characteristic, machine learning

## Abstract

To satisfy the preference of each driver, the development of a Lane-Keeping Assistance (LKA) system that can adapt to individual drivers has become a research hotspot in recent years. However, existing studies have mostly relied on the assumption that the LKA characteristic aligned with the driver’s preference is consistent with this driver’s naturalistic driving characteristic. Nevertheless, this assumption may not always hold true, causing limitations to the effectiveness of this method. This paper proposes a novel method for a Driver-Adaptive Lane-Keeping Assistance (DALKA) system based on drivers’ real preferences. First, metrics are extracted from collected naturalistic driving data using action point theory to describe drivers’ naturalistic driving characteristics. Then, the subjective and objective evaluation method is introduced to obtain the real preference of each test driver for the LKA system. Finally, machine learning methods are employed to train a model that relates naturalistic driving characteristics to the drivers’ real preferences, and the model-predicted preferences are integrated into the DALKA system. The developed DALKA system is then subjectively evaluated by the drivers. The results show that our DALKA system, developed using this method, can enhance or maintain the subjective evaluations of the LKA system for most drivers.

## 1. Introduction

Advanced Driver Assistance Systems (ADAS) are designed to enhance both driving safety and comfort. Lane-Keeping Assistance (LKA) is one type of ADAS that prevents the hazards resulting from unintended lane departure. However, during the design process of ADAS, insufficient consideration is given to the differences in preferences among drivers. Existing literature has shown that drivers of different genders, ages, and driving experiences have different levels of acceptance regarding ADAS [1,2].

One approach to addressing this issue is to offer mode selection for drivers. For instance, in the case of Adaptive Cruise Control (ACC), various time headway modes—such as short, normal, and long—could be made available, allowing drivers to choose the mode that best suits their preferences via the human–machine interface. Although this approach can help customize ADAS for individual drivers, there are also some potential challenges to consider. For instance, when a driver lacks enough ADAS experience, he may be unsure which mode would best satisfy his preference. On the other hand, the number of ADAS modes provided may be severely limited, which can restrict driver choice and make it difficult to find the most suitable mode. As a result, driver-adaptive ADAS that can automatically satisfy the preferences of different drivers has become a research hotspot in recent years.

The current primary approach for developing driver-adaptive ADAS is to learn and mimic the naturalistic driving characteristics of the current driver, aiming to make system characteristics satisfying the driver’s preference. Naturalistic driving characteristics are the behavior and performance exhibited during a driver’s manual driving process, i.e., when not using ADAS [3,4]. The development methods of driver-adaptive ADAS mainly include approaches based on classification characteristics and those based on individual characteristics. Classification-based methods involve categorizing drivers into different groups based on their naturalistic driving characteristics, followed by configuring distinct ADAS features for each group. Specifically, drivers are classified into categories such as “conservative”, “normal”, and “aggressive” based on metrics from driving data, such as lateral position and speed within the lane [5] and time to lane-crossing [6,7]. Some studies, not relying on metrics, directly employ non-parametric methods like Gaussian mixture models [8] for driver classification. After classification, personalized ADAS characteristics are tailored to different driving characteristic groups using methods like fuzzy rule tables [5] or averaging within-group driving data [8]. Individual-based methods involve configuring ADAS characteristics for each driver based on their individual driving characteristics. By utilizing parameters from models fitted during a driver’s car-following behavior [9], dynamic expected driving range during lateral driving [10], or parameters from Gaussian Mixture-Hidden Markov Models [11,12], personalized system characteristics for ACC, LKA, and other ADAS are set for each driver. These studies enable ADAS to effectively replicate the driver’s unique driving characteristic, thereby achieving differentiated ADAS.

However, whether the characteristics of the LKA system that are most preferred by the driver should align with his own naturalistic driving characteristics has become a key question. Some literature found that when a driver uses ADAS, the system characteristic he prefers may not be completely consistent with his own naturalistic driving characteristic. Some literature [13] compared the braking process of a driver during naturalistic driving and an autonomous driving vehicle when approaching a zebra crossing. It was observed that if the system characteristic was aligned with this driver’s naturalistic driving style, such a characteristic was often not rated as the safest and most cooperative by this driver. In the literature [14], which is focused on the overtaking process of autonomous driving vehicles, it was found that some drivers preferred the process that differed from their own driving styles. As for the LKA system, in our previous study [15], we utilized subjective and objective evaluation methods to quantify the differences in subjective ratings when drivers used an LKA system that aligned with their own driving characteristics compared to their real preferred LKA system. We found that among the 24 drivers participating in our test, 62.5% of them believed that the LKA system, which aligned with their individual driving characteristics, exhibited a significant difference compared to the LKA system they really preferred. Therefore, the method of aligning the characteristics of ADAS with the driver’s individual naturalistic driving characteristics may result in these characteristics not fully satisfying the drivers’ real preferences. In one study [16], an online personalized preference learning method was proposed based on driver preference feedback queries and Bayesian approaches, and it could quickly and accurately learn the preferences of most subjects. However, the driver preferences are assumed to be a simple linear function of some fixed driving characteristics, which may not be true.

In this paper, we focus on the LKA system and present a novel method for developing a Driver-Adaptive Lane-Keeping Assistance (DALKA) system. This method can be employed to initialize the driver preference model in the study [16], making the driver preference model closer to the real preferences of the drivers. Additionally, the method proposed in this paper can be applied in situations where online preference queries for drivers are not feasible. The main contributions of this paper are as follows:Extracting metrics for describing naturalistic driving characteristics based on action point theory (hereafter, these metrics will be referred to as “naturalistic driving characteristic metrics”);Introducing subjective and objective evaluation methods to obtain the test drivers’ real preferences to the LKA system, making model training possible;Instead of having the LKA system directly mimic the driver’s naturalistic driving characteristics, employing machine learning models to train a model using the driver’s individual driving characteristics and their real preferred LKA system characteristics and integrating the model-predicted drivers’ real preferences into the LKA system.

The remaining content of this paper is organized as follows. Section 2 introduces the development method of DALKA. Section 3 describes the experimental platform and process. Section 4 presents the drivers’ real preferences, which were used to train the model, and the predictive performance of the model. Section 5 explains how the predicted preferences are integrated into the LKA system, along with the results of the validation experiments. Section 6 gives a summary of the entire paper and potential issues for further research.

## 2. Methods

### 2.1. Research Roadmap

The implementation roadmap of the proposed DALKA system is illustrated in Figure 1. In this implementation path, we follow the approach of configuring the LKA system parameters based on the analysis of naturalistic driving data. However, to better align the system characteristics with drivers’ real preferences, we introduce the “Driver Preference Prediction Model (DPPM)” in the implementation roadmap. In the DALKA system described in this paper, we do not conduct research on the environment perception module. The key focus of this study will be on the naturalistic driving data analysis module, DPPMs, and the LKA decision and control module.

The development roadmap for the DPPMs is illustrated in Figure 2. To train the DPPMs, it is necessary to obtain the test drivers’ real preferences for the LKA system during the model training phase. The real preferences, along with the driver’s naturalistic driving characteristic metrics, are used as training samples to train the DPPM.

### 2.2. Lateral Naturalistic Driving Characteristic Analysis Method

Naturalistic driving characteristics are the driving behaviors and performance during the driver’s manual driving process (as stated before). It can provide an intuitive insight into the driving behavior of an individual driver [17]. Common analysis methods for naturalistic driving characteristics mainly include the descriptive statistics method [6,10], the parameter estimation method [18,19], and the non-parameter estimation method [20]. The descriptive statistics method refers to using basic statistical metrics such as the mean and standard deviation of various variables during a driver’s naturalistic driving process to describe each characteristic. This method is easy to apply, and the metrics have clear significance. However, the metrics extracted by this method heavily rely on experience, and overly simple statistics may struggle to precisely capture specific aspects of a driver’s driving characteristics. The parameter estimation method refers to initially describing the naturalistic driving process using a driver model with clear physical meanings. Subsequently, different model parameters are estimated for different drivers, serving as metrics to characterize distinct naturalistic driving characteristics. However, due to potential deficiencies in the driver model, its predictive accuracy may be compromised, leading to inaccuracy when describing the naturalistic driving process. On the other hand, the non-parameter estimation method aims to enhance model accuracy by using a black-box model to describe the naturalistic driving process and estimate model parameters. This method achieves higher predictive accuracy for driving processes but lacks clear physical meanings for model parameters, limiting its applicability in describing naturalistic driving characteristics. Considering these factors, we extend the traditional descriptive statistics method to overcome its limitations of single-dimensional metrics and shallow quantification of driving characteristics. By applying action point theory, we enhance the descriptive statistics method.

#### 2.2.1. Traditional Descriptive Statistics Method

In this study, we collected the lateral offset, steering wheel angle, steering wheel torque, yaw rate, and their first and second derivatives with respect to time as basic variables during the naturalistic driving process. Statistical metrics of these basic variables were computed for all of the driving data to serve as metrics of naturalistic driving. These metrics are categorized into three aspects: basic metrics, steering returning metrics, and frequency-domain metrics. Basic metrics include the mean, standard deviation, 5th percentile, and 95th percentile of basic variables. Steering returning metrics, based on the analysis in ref. [10], including steering returning frequency $f_{\theta \mathrm{st}-\mathrm{peak}}$ (the number of steering wheel angle peak points per unit time), lateral offset returning frequency $f_{\mathrm{ypeak}}$ (the number of lateral offset peak points per unit time), as well as the variance of lateral offset $s_{y-\mathrm{lane}-\mathrm{stpeak}}^{2}$ and lateral offset speed $s_{vy-\mathrm{lane}-\mathrm{stpeak}}^{2}$ at the steering wheel angle peak points. Frequency-domain metrics include $f_{\mathrm{FFT}-\theta \mathrm{st}}$ and $f_{\mathrm{FFT}-y}$, representing the frequencies corresponding to the amplitude peaks after Fourier-transforming the steering wheel angle and lateral offset, respectively.

#### 2.2.2. Descriptive Statistics Method Based on Action Point Theory

The action point theory was first proposed in the study of longitudinal car-following processes. In contrast to modeling the driver, the action point theory is based on the direct analysis of driving processes and driver control behaviors. Action points have a clear physical meaning, making it more straightforward to apply in the analysis of driving characteristics [21]. In ref. [22], the relationship between the relative longitudinal distance and relative velocity of two vehicles in the longitudinal car-following process is utilized to propose several key indicators reflecting action points, including SDV (the threshold of speed difference at large distances), CLDV (the threshold for recognizing small speed differences at short, decreasing distances), OPDV (the threshold for recognizing small speed differences at short, increasing distances), and AX (the desired distance between the front of successive vehicles in a standing queue). In ref. [23], considering the delay between driver-executed actions and the vehicle’s longitudinal speed response, the relationship graph between actuator signals (such as accelerator pedal pressure) and relative velocity is introduced, providing a closer approximation to the driver’s action points.

We apply action point theory in longitudinal driving to the lateral naturalistic driving process, extracting action points for the lane-keeping process based on the steering wheel angle and lateral offset. The specific extraction process is the same as the method described in ref. [15].

The three action points during the lane-keeping process in naturalistic driving are illustrated in Figure 3, specifically:Lane-Keeping Steering Starting Point, LKSSP:

The moment when the driver initiates steering to bring the vehicle back to the center of the lane, typically when perceiving a risk of deviating out of the lane;

Lane-Keeping Lateral Maximum Deviation Point, LKMDP:

The moment following LKSSP when the lateral offset of the vehicle reaches its peak. At this moment, the vehicle’ tendency to deviate toward outside of the lane is stopped, and the driver no longer perceives a risk of lane departure;

Lane-Keeping Steering Ending Point, LKSEP:

Lane-Keeping Steering Ending Point (LKSEP): The moment after LKMDP when the lateral offset returns to zero or when the velocity relative to the lane (referred to hereafter as “lane-relative velocity”) becomes zero. At this point, the driver steers the vehicle back to the lane center, marking the conclusion of one lane-keeping process.

**Figure 3 sensors-24-01666-f003:**
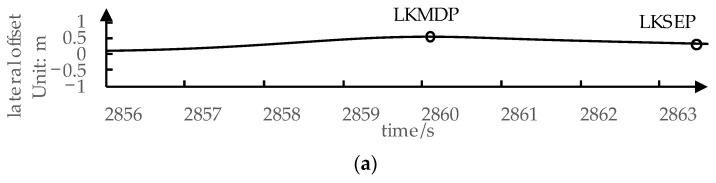
A piece of data of lane-keeping process and action points: (**a**) Lateral offset data; (**b**) Steering wheel angle data.

Based on these action points, we segmented naturalistic driving data to extract specific processes that better reflect lateral driving characteristics. The process between LKSSP and LKMDP is defined as the Risk-Perception Process. During this process, due to the continuous trend of the vehicle deviating from the lane, the driver focuses on perceiving the risk of lane departure. The process between LKSSP and LKSEP is defined as Returning Process. In Returning Process, the driver steers the wheel to correct the vehicle’s position back to the center of the lane. We also use lateral offset, steering wheel angle, steering wheel angular velocity, yaw rate, and their first and second derivatives as basic variables. The mean, standard deviation, 5th percentile, and 95th percentile of these variables are calculated as naturalistic driving characteristic metrics for Risk-Perception Process and Returning Process specifically.

Additionally, in ref. [15], based on the relationship between lateral offset and lane-relative velocity at LKSSP, parameters of a fitted line are used as metrics, reflecting the driver’s sensitivity to lane-relative velocity in risk perception. Furthermore, statistical metrics, including the mean, standard deviation, 50th percentile, and 95th percentile, are calculated for lateral offset and lane-relative velocity at LKSSP.

### 2.3. Method for Obtaining Drivers’ Real Preferences

In traditional design method of subjective and objective evaluation tests, whether based on system models or actual vehicles, diverse system characteristics for subjective evaluation (referred to as “evaluation samples”) are generated by altering internal system parameters. However, this approach is constrained by model or mechanical structure limitations, resulting in a limited scope covered by these characteristics. Therefore, we adopted the method used in ref. [24], which does not rely on a system model. Sample design metrics, which reflect system characteristics and have the potential to influence driver perception, are directly chosen. Subsequently, diverse evaluation samples are generated by varying the numerical values of each sample design metric. This method ensures a wide distribution of evaluation samples, contributing to an enhanced relationship between subjective and objective evaluations. In this section, we begin by analyzing the working process of LKA system. Subsequently, we present the method for constructing evaluation samples. Finally, we showcase the subjective evaluation questionnaire and objective metrics employed in the experiments.

#### 2.3.1. The Working Process of LKA System

In order to comprehensively describe the characteristics of the LKA system, we divide the working process of the LKA system into different sub-processes. When the vehicle gradually deviates from the lane and reaches a certain distance from the lane boundary, the LKA system intervenes based on certain intervention rules. It applies torque to the steering wheel to correct the vehicle back to the center of the lane.

Therefore, as shown in Figure 4, the LKA system’s working process can be divided into the following phases:Intervention timing: This refers to the situation at the moment when LKA system initiates its intervention $t_{\mathrm{LKA}-\mathrm{st}}$;Intervention process: This refers to the process from the moment when the LKA system initiates its intervention $t_{\mathrm{LKA}-\mathrm{st}}$ to the moment when the LKA system ends its intervention $t_{\mathrm{LKA}-\mathrm{ed}}$ due to the vehicle returning to the center of the lane.

**Figure 4 sensors-24-01666-f004:**
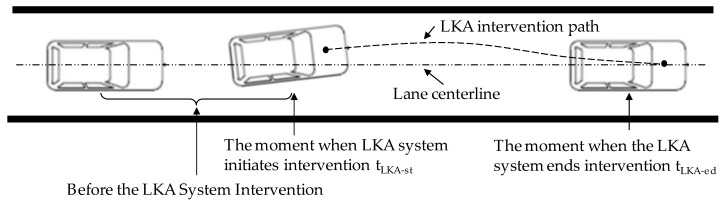
Illustrations of LKA intervention timing and LKA intervention process.

#### 2.3.2. Method for Constructing Evaluation Samples

The LKA intervention timing determines under what conditions the LKA system should start to intervene in the vehicle’s pose. The most common LKA intervention strategies are those based on the Distance to Lane Crossing (DLC) threshold [25]. Subsequently, strategies based on the Time to Lane Crossing (TLC) threshold were proposed to adapt to the different lane-relative velocities [26,27,28]. However, TLC-based strategies face challenges in situations where the vehicle is close to the lane boundary but parallel to the lane direction. To address this issue, combined strategies that use both DLC and TLC for intervention have been proposed [7,29]. Although different studies have adopted various design rules, they are generally based on DLC and TLC. Furthermore, TLC can be derived from DLC and the lateral velocity of the vehicle relative to the lane direction, which is denoted as the lane-relative velocity $v_{y-\mathrm{lane}}$. Therefore, we can select the DLC threshold and lane-relative velocity $v_{y-\mathrm{lane}}$ as sample design metrics for LKA intervention timing.

In the LKA intervention process, the system initially corrects the vehicle’s heading to make it parallel to the lane. At this point, the vehicle has no tendency to deviate further from the lane, and the lateral offset reaches maximum. Subsequently, the system controls the vehicle back to the center of the lane. Therefore, there are three key points in the LKA intervention path: the starting point $\left(x_{1},y_{1}\right)$, the point of maximum lateral deviation $\left(x_{2},y_{2}\right)$, and the endpoint $\left(x_{3},y_{3}\right)$. Once these three key points are determined, the intervention path is planned based on two Bezier curves, $P_{1}\left(t\right)$ and $P_{2}\left(t\right)$. The smooth cubic Bezier curve is generated by adding two additional curve control points $\left(x_{11},y_{11}\right)$, $\left(x_{12},y_{12}\right)$, $\left(x_{21},y_{21}\right)$, $\left(x_{22},y_{22}\right)$ between adjacent key points, as shown in Figure 5a.

However, representing the LKA intervention path with coordinates of these control points may not be intuitive. Therefore, we transformed the coordinates of these points shown in Figure 5a into the variables shown in Figure 5b, which have clearer physical meanings, as shown in Equation (1).

The variables in Equation (1) can be chosen as sample design metrics. However, before this, some variables that are not suitable as sample design metrics due to information redundancy or experimental constraints need to be excluded. We can transform the initial lateral offset $y_{0}$ and initial yaw angle $\varphi_{0}$ into the initial lane-relative velocity $v_{y-\mathrm{lane}}$.
(1)$$\left\{\begin{array}{c} x_{1}=0,y_{1}=y_{0}\\ x_{11}=d_{1}+\cos\left(\varphi_{0}\right),y_{11}=y_{0}+d_{1}sin(\varphi_{0})\\ x_{12}=dis_{1}-d_{1},y_{12}=y_{0}+y_{\mathrm{offset}}\\ x_{2}={\mathrm{dis}}_{1},y_{2}=y_{0}+y_{\mathrm{offset}}\\ x_{21}={\mathrm{dis}}_{1}+d_{1},y_{21}=y_{0}+y_{\mathrm{offset}}\\ x_{22}={\mathrm{dis}}_{2}-d_{2},y_{22}=0\\ x_{3}={\mathrm{dis}}_{2},y_{3}=0 \end{array}\right.$$

We can keep the total longitudinal distance dis of the LKA intervention process and exclude ${\mathrm{dis}}_{1}$ with high correlation, then convert $y_{\mathrm{offset}}$ (the maximum distance further deviating toward the outside of the lane relative to $y_{0}$) into r (the ratio between the minimum distance from the lane boundary during the intervention process ${\mathrm{DLC}}_{\min}$ and the distance from the lane boundary at LKA intervention ${\mathrm{DLC}}_{0}$), thus avoiding instances of deviating out of the lane in certain scenarios. The formula for calculating r is as shown in Equations (2) and (3).
(2)$$r={\mathrm{DLC}}_{\min}/{\mathrm{DLC}}_{0},$$
(3)$$\left\{\begin{array}{c} {\mathrm{DLC}}_{\min}=\frac{1}{2}\left(w_{\mathrm{lane}}-w_{\mathrm{vehicle}}-w_{\mathrm{mark}}\right)-y_{0}-y_{\mathrm{offset}}\\ {\mathrm{DLC}}_{0}=\frac{1}{2}\left(w_{\mathrm{lane}}-w_{\mathrm{vehicle}}-w_{\mathrm{mark}}\right)-y_{0}\;\;\;\;\;\;\;\;\;\;\;\;\;\;\; \end{array}\right.,$$

As shown in Figure 5b, in Equation (3), $w_{\mathrm{lane}}$ is the width of the lane, $w_{\mathrm{vehicle}}$ is the width of the vehicle, $w_{\mathrm{mark}}$ is the width of the lane mark, $y_{0}$ is the initial lateral offset of LKA intervention process, and $y_{\mathrm{offset}}$ is the maximum distance further deviating toward the outside of the lane relative to $y_{0}$.

Finally, we can eliminate the Bezier curve control arm lengths $d_{1}$ and $d_{2}$, which have little impact on the path shape. In conclusion, the sample design metrics for the LKA intervention process are as follows: the initial lane-relative velocity $v_{y-\mathrm{lane}}$, the total longitudinal distance dis, and the ratio r between ${\mathrm{DLC}}_{\min}$ and ${\mathrm{DLC}}_{0}$.

We employed the uniform design method in experimental design to achieve an even distribution of metrics across various samples. This method eliminates the necessity for numerous repetitive experiments and demonstrates a certain robustness to variations in the model [30]. The uniform experimental design table $U_{n}\left(q^{s}\right)$ contains approximate optimal combinations of metric values under different numbers of experiments n, values of metrics q, and quantity of metrics s [30]. We employed uniform experimental design tables $U_{9}\left(9^{2}\right)$ and $U_{9}\left(9^{3}\right)$ to derive the sample characteristics of the LKA system for intervention timing and intervention process, respectively, as shown in Table 1 and Table 2.

#### 2.3.3. The Subjective Questionnaires for LKA System and Objective Metrics

Relevant research has previously proposed subjective evaluation questions related to driver perception in LKA intervention timing [28,29] and LKA intervention process [31,32]. In this study, we integrated evaluation questions from these studies, eliminating redundancy to form a comprehensive subjective assessment questionnaire regarding the LKA intervention timing and the LKA intervention process. The subjective evaluation questions, scoring ranges, and optimal scores are outlined in Table 3.

Sample design metrics are only used for constructing evaluation samples. To establish a subjective and objective evaluation model, objective metrics still need to be extracted. The extracted objective metrics are presented in Table A1.

### 2.4. Method for Training Driver Preference Prediction Model

Utilizing natural driving metrics of drivers to predict their preferences for LKA system is fundamentally a regression problem. Random Forest (RF) is a Bagging-style ensemble learning method based on decision trees or regression trees. Given the difficulty of obtaining extensive experimental data through subjective evaluation tests, among various machine learning methods, RF stands out for its advantages in controlling model overfitting and requiring a smaller amount of data. Additionally, the method’s importance ranking based on node impurity provides excellent support for model analysis. Therefore, we choose RF method to train the model in predicting driver preferences, referred to as the Driver Preference Prediction Model (DPPM). The modeling approach of Random Forest can be referenced from [33].

## 3. Tests

### 3.1. Test Conditions and Procedure

We focused on the research of lateral natural driving characteristics and the LKA system. In order to eliminate the influence of different drivers’ longitudinal speed control abilities on their steering control during naturalistic driving and the perception of the LKA system, we ensured that drivers did not need to control the longitudinal speed during the experiments. A constant speed of 80 km/h was set for the experiments.

The procedure of the lane-keeping data-collection test is outlined in Table A2. The subjective evaluation tests were divided into three tests for each working process of the LKA system. The procedures for each test of LKA working processes are shown in Table A3.

### 3.2. Test Platform

Our experiments were conducted on a fixed-base driving simulator. It consists of three main components: a steering feedback simulation device consisting of a Steering-Force-Feedback Actuator (FFA) system, a rapid prototyping controller for vehicle dynamics, an EPS model, and LKA controller computations, as well as a computer with a screen for generating virtual reality environments and simulating traffic flow.

The overall architecture and a physical illustration of the driving simulator are shown in Figure 6.

### 3.3. Test Drivers

We recruited test drivers for naturalistic driving data-collection tests and subjective evaluation tests. These drivers had a certain experience and understanding of the LKA system. We primarily selected researchers with more than 3 years of driving experience engaged in relevant research projects and engineers from automotive companies. Driver information is shown in Table 4.

## 4. Result

### 4.1. Driver’s Real Preference for the LKA System

In this section, the results of drivers’ real preferences with regard to LKA intervention timing and the LKA intervention process will be presented. Firstly, we establish models comparing subjective evaluations and objective metrics. Subsequently, the obtained models are analyzed to identify the key metrics that influence drivers’ subjective evaluations. These metrics are applied to the subsequent LKA decision and control module. Finally, drivers’ real preferred values for these metrics can be obtained based on optimal subjective ratings.

#### 4.1.1. Driver’s Preference Regarding LKA Intervention Timing

For LKA intervention timing, a linear model can effectively represent the relationship between a driver’s subjective evaluation and objective metrics ${\mathrm{DLC}}_{0}$ and $v_{y-\mathrm{lane}0}$. The models of different drivers can all be uniformly expressed as Equation (4):(4)$$Q_{1}=\beta_{2}\times \;{\mathrm{DLC}}_{0}+\beta_{1}\times v_{y-\mathrm{lane}0}+\beta_{0}=\beta_{2}\left({\mathrm{DLC}}_{0}+\frac{\beta_{1}}{\beta_{2}}\times \;v_{y-\mathrm{lane}0}+\frac{\beta_{0}}{\beta_{2}}\right),$$

In the equation, ${\mathrm{DLC}}_{0}$ and $v_{y-\mathrm{lane}0}$ are the objective metrics, as shown in Table A1. The coefficients $\beta_{0}$, $\beta_{1}$, and $\beta_{2}$ are model parameters, which vary for different drivers and represent their preferences.

By setting $Q_{1}=0$ (which corresponds to the highest satisfaction rating given by the drivers, as shown in Table 3), we can derive the following equation:(5)$${\mathrm{DLC}}_{\mathrm{th}}=f_{\beta_{0},\beta_{1},\beta_{2}}\left(v_{y-\mathrm{lane}}\right)=-\frac{\beta_{1}}{\beta_{2}}\;\times \;v_{y-\mathrm{lane}}-\frac{\beta_{0}}{\beta_{2}}$$

From Equation (5), it can be found that the LKA intervention timing that the driver feels is most satisfactory is not a specific value of either ${\mathrm{DLC}}_{\mathrm{th}}$ or $v_{y-\mathrm{lane}}$ alone. Instead, it depends on the specific relationship between these two metrics, which is determined by the coefficients in Equation (5). Therefore, we introduce two new metrics to denote the key metrics that heavily influence drivers’ preferences: the virtual boundary offset distance ${\mathrm{offset}}_{VB}$ and the virtual boundary crossing time ${\mathrm{TLC}}_{VB}$, as shown in Equation (6):(6)$${\mathrm{offset}}_{\mathrm{VB}}=-\beta_{0}/\beta_{2},\;\;\;\;\;{\mathrm{TLC}}_{\mathrm{VB}}=-\beta_{1}/\beta_{2}$$

Equation (5) can be rewritten as
(7)$${\mathrm{DLC}}_{\mathrm{th}}=f_{{\mathrm{offset}}_{\mathrm{VB}},{\mathrm{TLC}}_{\mathrm{VB}}}\left(v_{y-\mathrm{lane}}\right)={\mathrm{TLC}}_{\mathrm{VB}}\;\times \;v_{y-\mathrm{lane}}+{\mathrm{offset}}_{\mathrm{VB}}$$

By setting $Q_{1}=0$, we can derive the preferred values of ${\mathrm{offset}}_{\mathrm{VB}}$ and ${\mathrm{TLC}}_{\mathrm{VB}}$ for 10 drivers, as shown in Table 5.

Combining Table 4, we can explore the relationship between age and drivers’ preferences for LKA intervention timing, as shown in Figure 7. It can be observed that although the relationship between age and preference is not very clear, drivers aged 30 and above tend to prefer a larger (i.e., safer) ${\mathrm{offset}}_{\mathrm{VB}}$, as indicated in Figure 7a. On the other hand, drivers in the age group of 24 to 26 tend to prefer a smaller (i.e., more aggressive) ${\mathrm{offset}}_{\mathrm{VB}}$. In this age group, only one driver prefers a larger ${\mathrm{offset}}_{\mathrm{VB}}$ (0.66 m), while the rest prefer an ${\mathrm{offset}}_{\mathrm{VB}}$ below 0.50 m. Regarding ${\mathrm{TLC}}_{\mathrm{VB}}$, drivers aged 30 and above tend to prefer a smaller ${\mathrm{TLC}}_{\mathrm{VB}}$, indicating that these drivers are not sensitive to lateral speed (i.e., they do not prefer LKA to intervene earlier as the deviation speed increases). This may be because these drivers prefer a larger ${\mathrm{offset}}_{\mathrm{VB}}$, indicating that their focus is more on the position of the vehicle deviating from the lane rather than the lane-relative velocity. However, overall, the relationship between drivers’ preferences for LKA intervention timing and age is not clear. It is challenging to predict drivers’ preferences just based on age.

#### 4.1.2. Driver’s Preference for LKA Intervention Process

Regarding subjective evaluation questions of the LKA intervention process $Q_{2}$ and $Q_{3}$, linear models do not yield satisfactory results. Therefore, we employed the RF method for modeling. The RF models for $Q_{2}$ and $Q_{3}$ achieve average Mean Absolute Error (MAE) values of 0.023 and 0.025 on the test sets, respectively.

During the training of the Random Forest, in the process of building each individual base regression tree, the impurity of each input (i.e., the objective metrics in this paper) is calculated. The objective metric with the lowest impurity at each node is selected for partitioning, resulting in the creation of new subsets for further splitting. Therefore, recording the impurity of nodes during the training process can serve as a basis for assessing the importance of each objective metric, allowing for their importance ranking and potential feature reduction [33]. In regression trees, node impurity is typically measured using the residual sum of squares (RSS):(8)$$\mathrm{RSS}=\sum_{\left(x_{i},y_{i}\right)\in D_{v1}}{\left(y_{i}-{\hat{y}}_{\mathrm{Dv}1}\right)}^{2}+\sum_{\left(x_{i},y_{i}\right)\in D_{v2}}{\left(y_{i}-{\hat{y}}_{\mathrm{Dv}2}\right)}^{2}$$
where $D_{v1}$ and $D_{v2}$ are the subsets formed by splitting the node data based on a certain criterion, and ${\hat{y}}_{\mathrm{Dv}1}$ represents the mean of the outputs $y_{i}$ in subset Dv1.

Based on the node impurity of objective metrics, the most important objective evaluation metrics can be identified. For $Q_{2}$, which concerns the driver’s perception of the vehicle’s motion when it returns to the center of the road, the four important objective evaluation metrics selected are $\omega_{r-\mathrm{mean}}$, $\theta_{\mathrm{st}-\mathrm{mean}}$, ${\mathrm{DLC}}_{\max}$, and $T_{p}$. For $Q_{3}$, which focuses on the driver’s perception of the minimum distance between the vehicle and the lane boundary, the three important objective evaluation metrics selected are ${\mathrm{DLC}}_{\min}$, ${\mathrm{DLC}}_{\mathrm{th}}$, and $v_{y-\mathrm{lane}-\mathrm{mean}}$.

The extraction of important objective evaluation metrics based on node impurity cannot avoid internal correlations among these metrics, leading to potential information redundancy. The correlations among these metrics were analyzed. The correlation coefficients between $\theta_{\mathrm{st}-\mathrm{mean}}$, ${\mathrm{DLC}}_{\max}$, and $T_{p}$ with $\omega_{r-\mathrm{mean}}$ are 1, −0.96, and −0.91, respectively. The correlation coefficients between ${\mathrm{DLC}}_{\mathrm{th}}$, $v_{y-\mathrm{lane}-\mathrm{mean}}$, and ${\mathrm{DLC}}_{\min}$ are 0.97 and −0.91, respectively. Consequently, the final set of retained metrics is $\omega_{r-\mathrm{mean}}$ and ${\mathrm{DLC}}_{\min}$.

We utilized a grid search to optimize the value of important objective metrics to obtain preferences. The objective metrics values for the LKA intervention process preferences of eight drivers are shown in Table 6.

Combining Table 4, we can also explore the relationship between age and drivers’ preferences for the LKA intervention process, as shown in Figure 8. There is almost no clear relationship between age and drivers’ preference for $\omega_{r-\mathrm{mean}}$. In Figure 8b, it can be observed that as drivers′ age increases, their preference for a ${\mathrm{DLC}}_{\min}$ tends to decrease. This indicates that younger drivers prefer to correct the vehicle′s heading more quickly, leading to a larger ${\mathrm{DLC}}_{\min}$, while older drivers are less inclined to conduct overly aggressive heading-correction maneuvers.

### 4.2. Predictive Performance of DPPM

We used 80% of the data from the dataset as the training set and the remaining 20% as the test set. The predicted values and actual values for ${\mathrm{offset}}_{\mathrm{VB}}$ by DPPM are compared in Figure 9. The MAE for the training set and the test set is 0.01 m and 0.09 m, respectively. The predicted values and actual values for ${\mathrm{TLC}}_{\mathrm{VB}}$ determined by DPPM are compared in Figure 10. The MAE for the training set and test set are both 0.01 m.

The predicted values and actual values for $\omega_{r-\mathrm{mean}}$ by DPPM are compared in Figure 11. The MAE for the training set and test set are 0.01 deg/s and 0.03 deg/s, respectively. The predicted values and actual values for ${\mathrm{DLC}}_{\min}$ determined by DPPM are compared in Figure 12. The MAE for the training set and test set are 0.01 m and 0.04 m, respectively.

### 4.3. Discussion of Results

Although the initial demonstration of the model’s predictive performance in Section 4.2 through MAE provides insights, there is still a lack of established indices for determining an appropriate level of accuracy. In this study, a greater deviation between the metric’s value predicted by DPPM and the actual value from drivers could result in lower subjective ratings for the DALKA system. This deviation may potentially extend beyond the acceptable range shown in Table 3. To address this, we introduced two indices: the tolerance $\Delta_{*}$ and the DPPM prediction accuracy ${\mathrm{Fit}}_{*}$. $\Delta_{*}$ represents the range of objective metric values for which the driver’s subjective ratings are in the acceptable range. ${\mathrm{Fit}}_{*}$ is the proportion of DPPM predictions with absolute errors smaller than $\Delta_{*}$ across all data samples, as expressed in Equation (9):(9)$${\mathrm{Fit}}_{*}=\frac{1}{n}\sum_{i=1}^{n}𝟙\left(\left|{\hat{y}}_{i}-y_{i}\right|\;<\;\Delta_{*}\right)$$n is the number of data samples. ${\hat{y}}_{i}$ is the value predicted by DPPM for a specific objective metric of the i-th data sample, while $y_{i}$ is the actual value. $\Delta_{*}$ is the tolerance for the metric. The function 𝟙() is an indicator function, yielding “1” when the condition inside the parentheses is true, and “0” otherwise.

According to Table 3, drivers are considered within an acceptable range when their subjective ratings fall within $\left[-1,1\right]$. Based on the main and objective evaluation models from Section 4.1, the ranges for input metrics can be determined. The average tolerances are as follows: ${\overset{-}{\Delta }}_{\mathrm{offsetVB}}=0.26$ m, ${\overset{-}{\Delta }}_{\mathrm{TLCVB}}=0.54$ m/s. The minimum tolerances are as follows: $\Delta_{\mathrm{offsetVB}}^{\min}=0.12$ m, $\Delta_{\mathrm{TLCVB}}^{\min}=0.3$ m/s.

As shown in Figure 13, for the prediction of ${\mathrm{offset}}_{\mathrm{VB}}$, DPPM achieves ${\mathrm{Fit}}_{\mathrm{offsetVB}}=92\%$ on the testing set under average tolerance ${\overset{-}{\Delta }}_{\mathrm{offsetVB}}$ and ${\mathrm{Fit}}_{\mathrm{offsetVB}}=70\%$ under the minimum tolerance $\Delta_{\mathrm{offsetVB}}^{\min}$. This implies that 92% of the ${\mathrm{offset}}_{\mathrm{VB}}$ values predicted by DPPM are within the acceptable range for drivers.

As shown in Figure 14, for the prediction of ${\mathrm{TLC}}_{\mathrm{VB}}$, DPPM achieves a ${\mathrm{Fit}}_{\mathrm{TLCVB}}$ of 100% on the testing set under both average tolerance ${\overset{-}{\Delta }}_{\mathrm{TLCVB}}$ and minimum tolerance $\Delta_{\mathrm{TLCVB}}^{\min}$. This implies that all the predicted values of ${\mathrm{TLC}}_{\mathrm{VB}}$ by DPPM fall in the acceptable range of drivers.

For subjective and objective evaluation models of the LKA intervention process, trained using RF models, predictions for various inputs are obtained by traversing the input space. This process allows us to determine the input ranges corresponding to outputs within the $\left[-1,1\right]$ range. The average tolerances are as follows: ${\overset{-}{\Delta }}_{\omega \mathrm{mean}}=0.23$ deg/s and ${\overset{-}{\Delta }}_{\mathrm{DLC}\min}=0.39$ The minimum tolerances are as follows: $\Delta_{\omega \mathrm{mean}}^{\min}=0.08$ deg/s and $\Delta_{\mathrm{DLCmin}}^{\min}=0.05$.

As shown in Figure 15, for the prediction of $\omega_{r-\mathrm{mean}}$, DPPM achieves ${\mathrm{Fit}}_{\omega rmean}=100\%$ on the testing set under average tolerance ${\overset{-}{\Delta }}_{\omega \mathrm{mean}}$ and ${\mathrm{Fit}}_{\omega rmean}=85\%$ under the minimum tolerance $\Delta_{\omega \mathrm{mean}}^{\min}$. This implies that all the predicted values for $\omega_{r-\mathrm{mean}}$ by DPPM fall in the acceptable range for drivers.

As shown in Figure 16, for the prediction of ${\mathrm{DLC}}_{\min}$, DPPM achieves ${\mathrm{Fit}}_{\mathrm{DLCmin}}=100\%$ on the testing set under average tolerance ${\overset{-}{\Delta }}_{\mathrm{DLC}\min}$ and ${\mathrm{Fit}}_{\mathrm{DLCmin}}=82.5\%$ under the minimum tolerance $\Delta_{\mathrm{DLCmin}}^{\min}$. This implies that the predicted values for ${\mathrm{DLC}}_{\min}$ are all within the acceptable range for drivers.

## 5. System Integration and Validation Test of DALKA

### 5.1. LKA Decision and Control Module

The LKA decision and control module consist of state decision module, path planning and control module, and output torque decision module, as shown in Figure 17.

We have extracted key metrics influencing drivers’ preferences: ${\mathrm{offset}}_{\mathrm{VB}}$, ${\mathrm{TLC}}_{\mathrm{VB}}$, $\omega_{r-\mathrm{mean}}$, and ${\mathrm{DLC}}_{\min}$. Once DPPM predicts values for these metrics, they are handed over to the LKA decision and control module for implementation. Therefore, to ensure that the LKA system meets the specified metric value, customization of the LKA decision and control module was undertaken.

The decision logic is illustrated in Figure 18.

In the state decision module, a new variable is introduced, which is the steering assistance torque gain coefficient α, used to represent whether the LKA system intervenes in control. The decision logic in detail is as follows:

Initially, the system receives the LKA system switch signal from the human–machine control panel. If the driver deactivates the LKA system, the system enters the off state, setting α to 0;

When the system confirms that the driver has activated the LKA system, it receives the status “If at least one lane line can be effectively detected” from the environment-perception module. If the status is “No,” indicating insufficient conditions for activating the LKA system, the system again enters the off state with α set to 0;If the environment-perception module confirms effective lane line detection, it evaluates the risk of the vehicle deviating from the lane by checking if the current DLC satisfies Equation (10):(10)$$\mathrm{DLC}<{\mathrm{DLC}}_{\mathrm{th}}$$Here, ${\mathrm{DLC}}_{\mathrm{th}}$ is the LKA intervention control threshold, calculated as Equation (11):(11)$${\mathrm{DLC}}_{\mathrm{th}}={\mathrm{TLC}}_{\mathrm{VB}}\times v_{y-\mathrm{lane}}+{\mathrm{offset}}_{\mathrm{VB}}$$${\mathrm{TLC}}_{\mathrm{VB}}$ and ${\mathrm{offset}}_{\mathrm{VB}}$ are the key metrics obtained from Section 4.1, influencing drivers’ preferences for LKA intervention timing, computed using DPPM. If Equation (10) is not met, the LKA system remains standby with α set to 0;If Equation (10) is satisfied, it is necessary to determine whether the driver has the intention of actively steering. We adopted the method proposed in refs. [34,35] to judge the driver’s intention to steer actively based on the steering wheel torque threshold $T_{\mathrm{st}}$, as shown in Equation (12). If Equation (12) is not satisfied, α is set to 0. Otherwise, the LKA system initiates its intervention, and α is set to 1.



(12)
$$T_{\mathrm{st}}<T_{\mathrm{st}-\max}$$



Regarding the path planning and control module, the path-planning method uses the same approach described in Section 2.3.1 when constructing the characteristics of the LKA intervention process. Regarding path-tracking control, numerous scholars have conducted research. Common methods include Linear Quadratic Regulator (LQR) control [36], sliding mode control [37], robust control [38], and model predictive control [39,40]. One study [41], considering the roll dynamics and network-induced delays, proposed a new multi-input, multi-output linear parameter-varying controller for path-tracking control. Another study [42] proposed a strategy based on the path-tracking preview algorithm and the LQR controller to improve the lateral stability of the vehicle and address the crosswind issue during driving. Compared to the above methods, sliding mode control is not only simple to implement but also robust to external disturbances. Yet another study [43] combined feedforward control based on the preview model and sliding mode control, which could control the maximum tracking error of the vehicle on the simulator within 0.1 m. Considering that the experiments in this study were conducted on a simulator with minimal external disturbances, we adopted the method used in ref. [43].

Regarding the logic in the output torque decision module, assuming the current state is at step k, the torque-control module receives α from the state-decision module and checks if Equation (13) is satisfied.
(13)α≥0.5

If Equation (13) is met at step k, indicating that the LKA system should be in the intervention control state with α=1 at the current step, $T_{\mathrm{LKout}}^{k}=T_{\mathrm{LKin}}^{k}$. In this case, the LKA torque $T_{\mathrm{LKA}}^{k}$ at step k is determined, as shown in Equation (14). The final output torque $T_{\mathrm{LKA}}$ of the LKA system at this point equals the lane-keeping torque $T_{\mathrm{LKin}}$ received from the path-planning and tracking module.
(14)$$T_{\mathrm{LKA}}^{k}=T_{\mathrm{LKout}}^{k}\cdot \alpha =T_{\mathrm{LKout}}^{k}=T_{\mathrm{LKin}}^{k}$$

If Equation (13) is not met at step k but was met at the previous step k − 1, it implies that the LKA system has just transitioned from the intervention control state to standby or off state at the current step. In this case, α=0, and $T_{\mathrm{LKout}}^{k}=T_{\mathrm{LKout}}^{k-1}$, meaning that for step k, k+1, and subsequent steps, $T_{\mathrm{LKout}}$ remains constant and equal to $T_{\mathrm{LKout}}^{k-1}$, as shown in Equation (15). The LKA torque $T_{\mathrm{LKA}}^{k}$ is determined, as shown in Equation (16).
(15)$$T_{\mathrm{LKout}}^{k-1}=T_{\mathrm{LKout}}^{k}=T_{\mathrm{LKout}}^{k+1}=T_{\mathrm{LKout}}^{k+2}=\ldots$$
(16)$$T_{\mathrm{LKA}}^{k}=T_{\mathrm{LKout}}^{k}\cdot \alpha =T_{\mathrm{LKout}}^{k-1}\cdot \alpha$$

Furthermore, the slope-constraint module restricts the rate of change of α, thereby preventing rapid withdrawal of the LKA torque, which could result in excessive steering by the driver.

### 5.2. Validation Test of DALKA System

To validate the effectiveness of the DALKA system, an additional subjective evaluation test was conducted by inviting 12 drivers who had not participated in the previous subjective and objective evaluation experiments. A comprehensive subjective evaluation was used to assess the overall performance of the integrated DALKA system. Ratings were given on a scale of $\left[1,5\right]$, with an acceptable range of $\left[4,5\right]$. For comparison, the drivers were also asked to provide a comprehensive evaluation of a fixed-characteristic LKA system. The key metric values for the fixed-characteristic LKA system were averaged based on the preferences of drivers, as obtained in Section 4.1. The result is shown in Table 7, and the comparison of the drivers’ evaluation of these systems is illustrated in Figure 19.

When using the DALKA LKA system, drivers gave an average subjective rating of 4.56, compared to 4.40 when using the fixed-characteristic LKA system. Regarding the acceptance of the drivers, it can be found that when using the LKA system with averagely preferred characteristic, 10 out of 12 drivers (83%) gave subjective ratings within the acceptable range. After experiencing the DALKA system, six drivers showed an improvement in subjective evaluations, three drivers maintained their subjective evaluations, and three drivers experienced a slight decline. However, the subjective evaluations of these three drivers remained within the acceptable range. In summary, subjective evaluations for the DALKA system from all 12 drivers (100%) fell within the acceptable range.

It can be observed that the DALKA system we developed demonstrates more pronounced adaptive effects for those drivers whose preference deviate significantly from the average preference. However, for drivers whose preferences align closely with the average preference, the DALKA system may lead to a decrease in subjective evaluation. Nonetheless, as these drivers already give high subjective evaluations for the average preference characteristics, it does not result in their evaluations falling outside the acceptable range.

## 6. Conclusions

Fixed, singular LKA system characteristics struggle to satisfy various preferences of different drivers. The DALKA system, which mimics drivers’ individual driving characteristics, addresses the issue of not meeting their real preferences.

The methodology presented in this paper, based on subjective and objective evaluations, provides a novel method of DALKA system development. Firstly, driver preferences to various LKA system processes are obtained through subjective and objective evaluation tests. Secondly, naturalistic driving characteristics are analyzed using action point theory to effectively describe individual lateral driving characteristics. Finally, DPPMs are built using the Random Forest method to predict LKA system characteristics preferred by drivers. The results show that, for ${\mathrm{offset}}_{\mathrm{VB}}$, 92% of the predicted values by DPPM are within the acceptable range for drivers. For ${\mathrm{TLC}}_{\mathrm{VB}}$, $\omega_{r-\mathrm{mean}}$ and ${\mathrm{DLC}}_{\min}$, all the predicted values for by DPPM fall in the acceptable range for drivers. Also, a validation test was conducted to verify the performance of DALKA system. It shows that when using DALKA system based on DPPM, the ratio of the drivers who think it is acceptable increases from 83% to 100%. The DALKA system show significant increases in subjective evaluation for those drivers whose preference deviate significantly from the average preference.

However, several limitations need to be addressed. The DALKA system development and related experiments in this paper are based on straight-road conditions. Further research is needed to explore other conditions such as curves. Additionally, while the DALKA system designed based on the DPPM model improves the acceptance of the LKA system for most drivers, there is a small group of drivers with reduced evaluation. For these drivers, it is necessary to explore more factors influencing driver preferences regarding the LKA system.

## Figures and Tables

**Figure 1 sensors-24-01666-f001:**
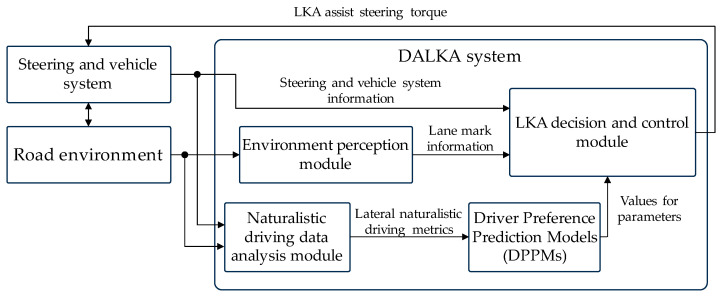
The implementation roadmap of the DALKA system.

**Figure 2 sensors-24-01666-f002:**
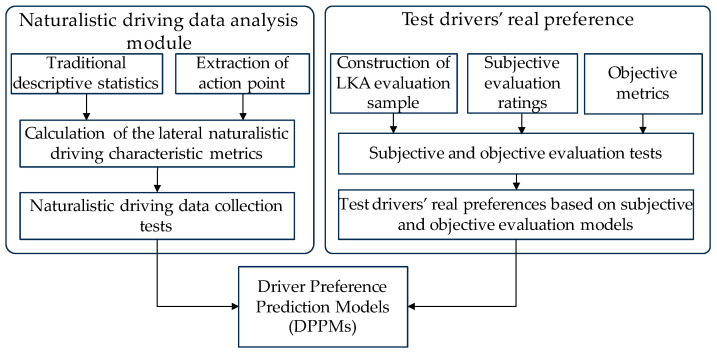
The development roadmap for the DPPMs.

**Figure 5 sensors-24-01666-f005:**
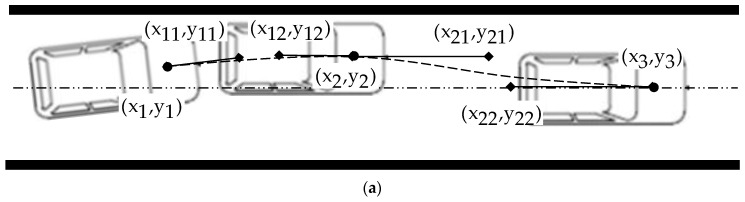
Path of LKA intervention process: (**a**) Key points and Bezier curve control points; (**b**) Objective metrics of path.

**Figure 6 sensors-24-01666-f006:**
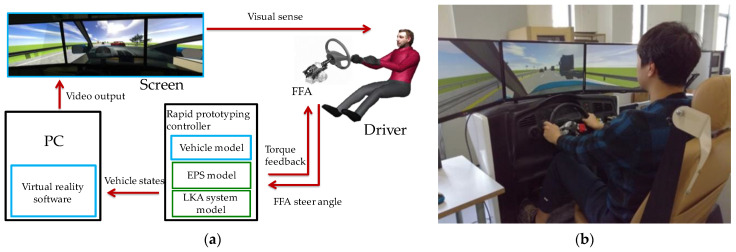
The driving simulator: (**a**) The overall architecture; (**b**) The physical illustration.

**Figure 7 sensors-24-01666-f007:**
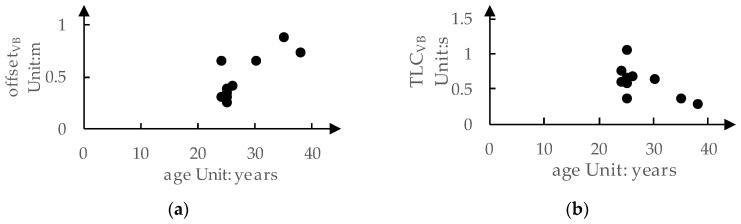
The relationship between drivers’ ages and their preference for LKA intervention timing: (**a**) ${\mathrm{offset}}_{\mathrm{VB}}$; (**b**) ${\mathrm{TLC}}_{\mathrm{VB}}$.

**Figure 8 sensors-24-01666-f008:**
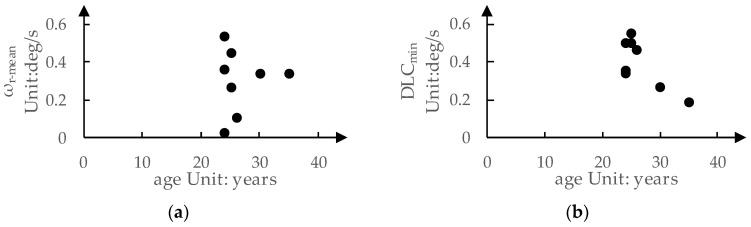
The relationship between drivers’ age and their preferences for LKA intervention process: (**a**) $\omega_{r-\mathrm{mean}}$; (**b**) ${\mathrm{DLC}}_{\min}$.

**Figure 9 sensors-24-01666-f009:**
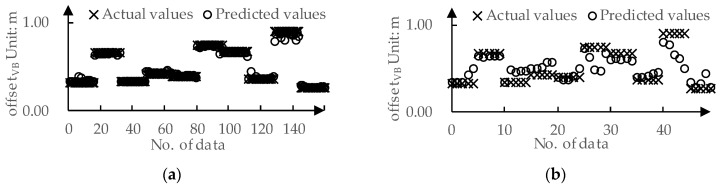
The predicted values and actual values for ${\mathrm{offset}}_{\mathrm{VB}}$: (**a**) Training set; (**b**) Testing set.

**Figure 10 sensors-24-01666-f010:**
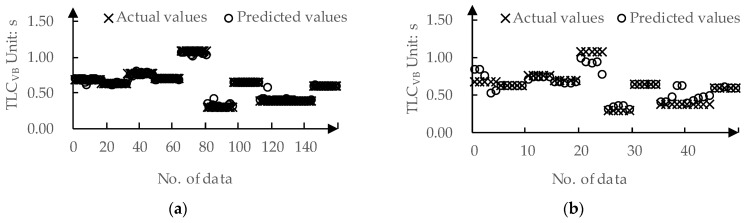
The predicted values and actual values for ${\mathrm{TLC}}_{\mathrm{VB}}$: (**a**) Training set; (**b**) Testing set.

**Figure 11 sensors-24-01666-f011:**
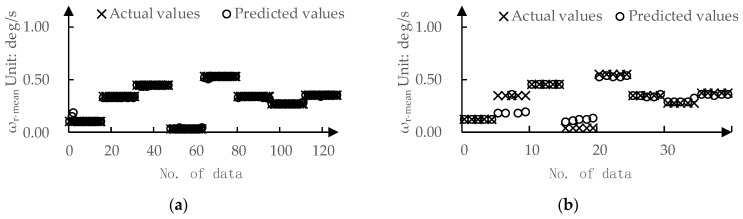
The predicted values and actual values for $\omega_{r-\mathrm{mean}}$: (**a**) Training set; (**b**) Testing set.

**Figure 12 sensors-24-01666-f012:**
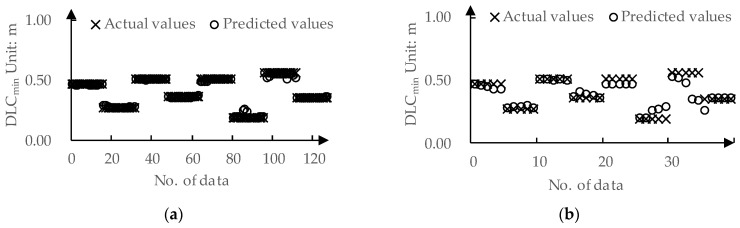
The predicted values and actual values for ${\mathrm{DLC}}_{\min}$: (**a**) Training set; (**b**) Testing set.

**Figure 13 sensors-24-01666-f013:**
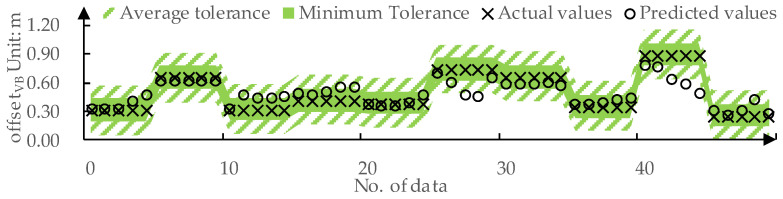
DPPM’s predicted values for ${\mathrm{offset}}_{\mathrm{VB}}$ on the test set compared to the tolerance.

**Figure 14 sensors-24-01666-f014:**
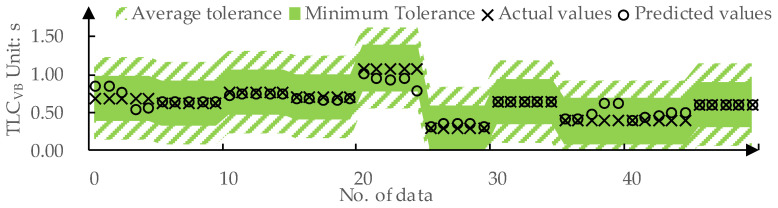
DPPM’s predicted values for ${\mathrm{TLC}}_{\mathrm{VB}}$ on the test set compared to the tolerance.

**Figure 15 sensors-24-01666-f015:**
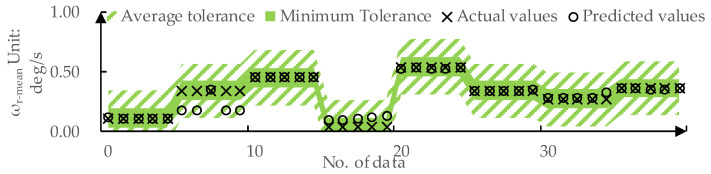
DPPM’s predicted values for $\omega_{r-\mathrm{mean}}$ on the test set compared to the tolerance.

**Figure 16 sensors-24-01666-f016:**
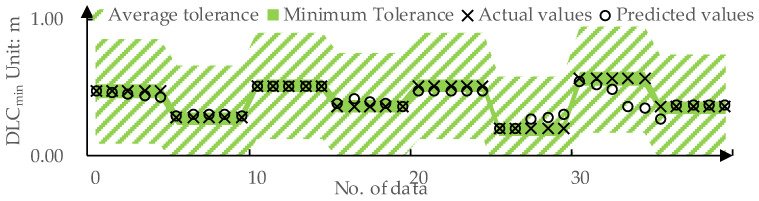
DPPM’s predicted values for ${\mathrm{DLC}}_{\min}$ on the test set compared to the tolerance.

**Figure 17 sensors-24-01666-f017:**
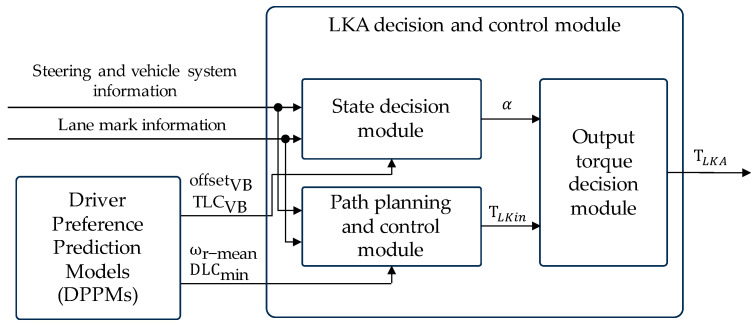
LKA decision and control module.

**Figure 18 sensors-24-01666-f018:**
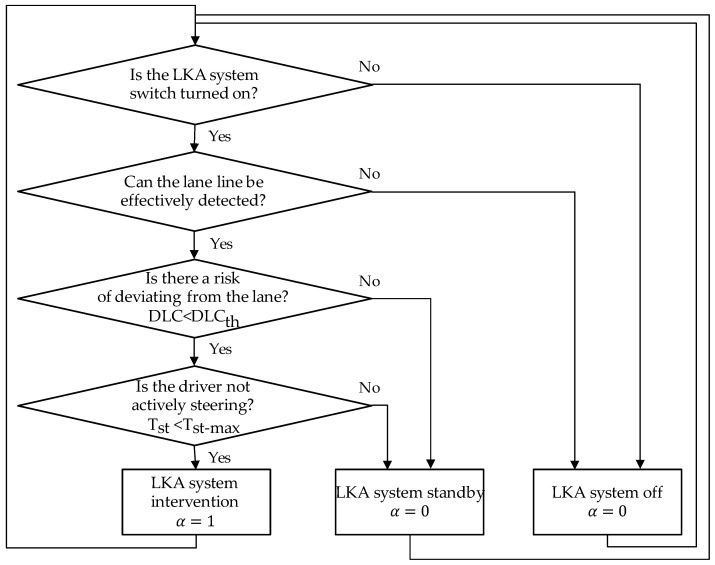
The decision logic for LKA system.

**Figure 19 sensors-24-01666-f019:**
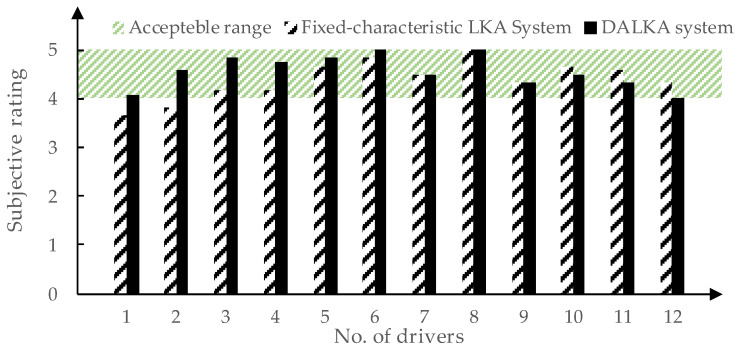
Comparison of drivers’ subjective evaluations of fixed-characteristic LKA system and DALKA system.

**Table 1 sensors-24-01666-t001:** Value of metrics under different evaluation samples for LKA intervention timing.

No.	${\mathrm{DLC}}_{\mathrm{th}}$ Unit: m	$v_{y-\mathrm{lane}}$ Unit: m/s
1	0.0	0.15
2	0.1	0.10
3	0.2	0.05
4	0.3	0.20
5	0.4	0.45
6	0.5	0.35
7	0.6	0.25
8	0.7	0.50
9	0.8	0.40
10	0.9	0.30

**Table 2 sensors-24-01666-t002:** Value of the metrics under different evaluation samples for LKA intervention process.

No.	$v_{y-\mathrm{lane}}$ Unit: m/s	dis Unit: m	r
1	0.20	90	0.3
2	0.35	85	0.7
3	0.50	80	0.2
4	0.15	75	0.6
5	0.30	70	0.1
6	0.45	65	0.5
7	0.10	60	0.0
8	0.25	55	0.4
9	0.40	50	0.8

**Table 3 sensors-24-01666-t003:** The subjective evaluation questions, scoring range, and optimal scores for LKA system.

Category	Subjective Evaluation Question	Scoring Range	Acceptable Range	Optimal Score
LKA intervention timing	$Q_{1}$: Is the intervention timing acceptable?	[−4,4]	[−1,1]	0
LKA intervention process	$Q_{2}$: Is the process of vehicle returning to road center acceptable?	[−4,4]	[−1,1]	0
	$Q_{3}$: During the intervention, is the minimum distance to lane line acceptable?	[−4,4]	[−1,1]	0

**Table 4 sensors-24-01666-t004:** Driver information.

No. of Driver	Job	Age Unit: Years	Driving Experience Unit: Years
1	Researcher	25	4
2	Researcher	24	5
3	Researcher	24	4
4	Researcher	26	4
5	Researcher	25	5
6	Engineer	38	10
7	Engineer	30	3
8	Other	25	3
9	Other	35	8
10	Engineer	25	4
11	Researcher	24	3

**Table 5 sensors-24-01666-t005:** Driver’s preferences for LKA intervention timing.

No. of Driver	${\mathrm{Offset}}_{\mathrm{VB}}$ Unit: m	${\mathrm{TLC}}_{\mathrm{VB}}$ Unit: s
1	0.31	0.68
2	0.66	0.62
3	0.32	0.77
4	0.42	0.70
5	0.39	1.08
6	0.74	0.30
7	0.66	0.65
8	0.35	0.39
9	0.89	0.39
10	0.26	0.60

**Table 6 sensors-24-01666-t006:** Driver’s preferences for LKA intervention process.

No. of Driver	$\omega_{r-\mathrm{mean}}$ Unit: deg/s	${\mathrm{DLC}}_{\min}$ Unit: m
1	0.45	0.51
2	0.54	0.51
3	0.03	0.36
4	0.11	0.47
5	0.27	0.56
7	0.34	0.27
9	0.34	0.19
11	0.36	0.35

**Table 7 sensors-24-01666-t007:** Drivers’ subjective evaluations of fixed-characteristic LKA system and DALKA system.

No. of Driver	Subjective Ratings of Fixed-Characteristic LKA System Unit: -	Does the Driver Find the Fixed-Characteristic LKA System Acceptable?	Subjective Ratings of DALKA System Unit: -	Does the Driver Find the DALKA System Acceptable?
1	3.67	No	4.08	Yes
2	3.83	No	4.58	Yes
3	4.17	Yes	4.83	Yes
4	4.17	Yes	4.75	Yes
5	4.67	Yes	4.83	Yes
6	4.83	Yes	5.00	Yes
7	4.50	Yes	4.50	Yes
8	5.00	Yes	5.00	Yes
9	4.33	Yes	4.33	Yes
10	4.67	Yes	4.50	Yes
11	4.58	Yes	4.33	Yes
12	4.33	Yes	4.00	Yes
Average	4.40	83%	4.56	100%

## Data Availability

Data are contained within the article.

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
