# Peer review of "A Method to Develop the Driver-Adaptive Lane-Keeping Assistance System Based on Real Driver Preferences"

_sensors, 2024, doi:10.3390/s24051666_

Round 1
Reviewer 1 Report
Comments and Suggestions for Authors
The document presents a new method for a Driver Adaptive Lane Keeping Assistance system that is based on driver preferences. Overall, the work appears interesting and it seems that the authors have put a lot of effort into it.
To enhance the manuscript's quality, please take into account the following suggestions.
1. Please include some references related to path tracking control, which is strongly linked to LKA systems. Consider recent works such as https://doi.org/10.1109/TITS.2023.3321415, https://doi.org/10.3390/app13179891.
2. It has concerns about the manuscript's formatting.The resolution of every figure and the quality of the equations need improvement.
3. The manuscript should include the gender of the participants during the experiment and a discussion on its relevance.
4. There are several typographical errors in the manuscript. Please check line 446.
5. Please expand the conclusions section by including some numerical indicators.
6. There are several references to a Table that does not exist. Please check Table XX on Table A3.
The English of the document seems adequate. It is recommended that some typos be revised.
Author Response
Thank you very much for your review and advice! Please see the attachment about the reply.

Reviewer 2 Report
Comments and Suggestions for Authors
The authors introduce an innovative approach for lane-keeping assistance tailored to individual driver preferences. The proposed lane-keeping assistance system, DALKA, integrates drivers' inherent driving characteristics with their stated preferences. The motivation behind this research stems from the observed disparity between drivers' natural driving tendencies and their expressed driving preferences.
However, the manuscript's presentation is deemed intricate and excessively lengthy. Moreover, the language employed is notably deficient, marked by numerous grammatical inaccuracies that signify imprecise writing. A comprehensive revision is recommended, with specific instances highlighted below:
- Line 31: "experiences" should replace "experience."
- Line 57: Correct the phrase to "Individual-based methods involve configuring ADAS characteristics for each driver based on their individual driving characteristics."
- Line 67: The term "literatures" should be corrected to "literature."
- Line 166: Ensure consistent notation for variance and speed.
- Line 167: Use "include" instead of "includes."
- Line 178: Clarify the acronyms SDV, CLDV, OPDV, and AX.
- Lines 256-257: Clarify the meaning of the sentence and the relationship between DLC, TLC, and literature.
Figure-related concerns:
- Figures 1, 15: Address arrows pointing to undefined locations.
- Figure 5: Enlarge for improved readability.
- Figure 16: Requires editing for clarity.
Equation-related issues:
- Equation 3: Define variables wlane, wvehicle, and wmark explicitly, as they are depicted in Figure 5 but lack textual definition.
General manuscript improvements:
- Correct inconsistent table numbering in the appendix.
- In Table 4, include information about test drivers' prior experience with LKA-equipped vehicles to account for potential biases in their subjective opinions.
- Expand the number of subject drivers if feasible, considering the potential impact of age as a distinguishing factor in the study.
- In Figure 17 and Section 5.2, provide a more detailed analysis of subjective ratings. Consider adding a table for precise and average results, enhancing clarity and ease of evaluation.
In summary, a thorough revision is necessary to rectify language-related issues, address acronyms, clarify unclear statements, and enhance overall manuscript coherence. Additionally, expanding and detailing certain sections would contribute to a more comprehensive and scientifically robust presentation.
Comments on the Quality of English LanguageA thorough revision is necessary to rectify language-related issues.
Author Response
Thank you very much for your review and advice. Please see the attachment for the reply and revision.

Round 2
Reviewer 1 Report
Comments and Suggestions for Authors
The authors have modified the work accordingly following the recommendations.
Reviewer 2 Report
Comments and Suggestions for Authors
The authors appear to have carefully addressed my concerns. While there may be room for further improvement, the paper is suitable for publication in its current form.